

# Effects of nettle slurry (*Urtica dioica* L.) used as foliar fertilizer on potato (*Solanum tuberosum* L.) yield and plant growth

Alfonso Garmendia[1], María Dolores Raigón[2], Olmo Marques[3], María Ferriol[1], Jorge Royo[3] and Hugo Merle[3]

[1] Instituto Agroforestal Mediterráneo, Universitat Politècnica de València, Valencia, Spain
[2] Instituto Universitario de Conservación y Mejora de la Agrodiversidad Valenciana, Universitat Politècnica de València, Valencia, Spain
[3] Departamento de Ecosistemas Agroforestales, Universitat Politècnica de València, València, Spain

## ABSTRACT

Organic agriculture is becoming increasingly important, and many natural products are now available for organic farmers to manage and improve their crops. Several ethnobotanical studies have indicated that the use of nettle slurry as fertilizer in organic farming for horticultural crops is spreading. Sometimes, however, the consequences of using these natural products have been poorly evaluated, and there is very little scientific evidence for the effects of using these slurries. In this study, we aimed to analyze the possible effect of nettle slurry on potato yields produced by organic farming. To achieve this main objective, we assessed the effect of nettle slurry on potato yields, plant size and growth parameters, chlorophyll content, and the presence of pests and diseases. Different slurry doses were assessed in 36 plots and nine variables were measured during the crop cycle. Under these field experimental conditions, nettle slurry (including one treatment with *Urtica* in combination with *Equisetum*) had no significant effects on yield, chlorophyll content, or the presence of pests and diseases in organic potato crops. The highest chlorophyll content was found in the double dose treatment, but the difference was not significant. This result, together with a small improvement in plant height with the double dose treatment, might indicate very slight crop enhancement which, under our experimental conditions, was not enough to improve yield. The *Urtica* and *Equisetum* slurry chemical analyses showed very low levels of nitrogen, phosphorus, and potassium.

Corresponding author
Hugo Merle,
humerfa@upvnet.upv.es

## INTRODUCTION

The latest survey on certified organic agriculture worldwide has shown that 50.9 million hectares of agricultural land were managed organically at the end of 2015 (*Lernoud & Willer, 2017*). All-important indicators have been increasing in the last decades: area, producers, and retail sales (*Lernoud & Willer, 2017*). This significant growth in organic farming is not only a matter of a marginal agricultural change, but also represents

the implementation of major changes in society, and their relation with agriculture (*Michelsen, 2001*; *Lobley, Butler & Reed, 2009*; *Reganold & Wachter, 2016*). Organic food consumption is associated with health beliefs and subjective well-being, which involves higher market values and demand (*Apaolaza et al., 2018*). Recent studies highlight that organic food is related with important benefits for human and environmental health (*Kahl & Rembiałkowska, 2014*; *Gomiero, 2017*). Moreover, in the next few years, agriculture will be pushed to become more sustainable as a global response to climate change.

Increasingly more natural products are available for organic farmers to manage and improve their crops (*Benfatto et al., 2015*). Many agrochemical companies are including organic fertilizers, natural herbicides or bio-based liquid formulations to control pests, in the products they offer. Some of these companies merely focus on natural products for organic farmers. However, the consequences of using these natural products for crop yields and other agro-ecosystem services have sometimes been poorly evaluated (*Gagic et al., 2017*).

Several ethnobotanical studies have indicated that the use of nettle (*Urtica dioica* L.) slurry as fertilizer in organic farming for horticultural crops is spreading in Spain (*Latorre, 2008*; *Benítez Cruz, 2009*). Small farmers can produce their own slurry, but most professional organic farmers usually buy the commercial product (field surveys). Companies must follow EC regulation No. 1107/2009 to obtain active substances. The purpose of this regulation is to ensure a high level of protection of both humans and the environment and, at the same time, to safeguard the competitiveness of the agriculture community (*European Union, 2009*). Many trading houses from different countries commercialize nettle slurry (e.g.: I.L.A.G.A., ARIES biogarten, General Organics, Trabe S. A., Asocoa, AgroBeta, MGI Developpement, etc.). Thus, nettle slurry is commonly used in organic agriculture and has a growing economic impact. Labeled general effects include leaf fertilizer, growth stimulator, ferric chlorosis control, pest and disease prevention, and insect repellents. Trading houses recommend three applications at 10% v/v and 300 l/ha over the crop cycle.

The chemical composition of nettle plants has been widely studied for medical purposes (*Zeković et al., 2017*). Its antioxidant capacity, therapeutic effect or immunological responses have often been recorded (*Buenz et al., 2017*; *Branisa et al., 2017*; *Saeidi Asl et al., 2017*). The potential industrial uses of stinging nettle were summarized by *Di Virgilio et al. (2015)*. They concluded that nettle have promising application in the food/feed, medicinal, and cosmetic sectors (*Di Virgilio et al., 2015*). Surprisingly, very few studies have centered on the agronomic use of nettle slurry as fertilizer, the chemical composition of fermented slurry, and its effect on crop yields. *Bozsik (1996)* carried out studies on the aphicidal efficiency of different nettle extracts. The cold-water extract had no significant effect on *Hyalopterus pruni* Geoffroy and *Cryptomyzus ribis* Linnaeus; the fermenting extract hardly influenced *H. pruni* and no significant efficiency was observed against *Aphis spiraephaga* F.P. Müller (*Bozsik, 1996*). *Rosnitschek-Schimmel (1985)* showed that the most important nitrogen compound in nettle plants was free

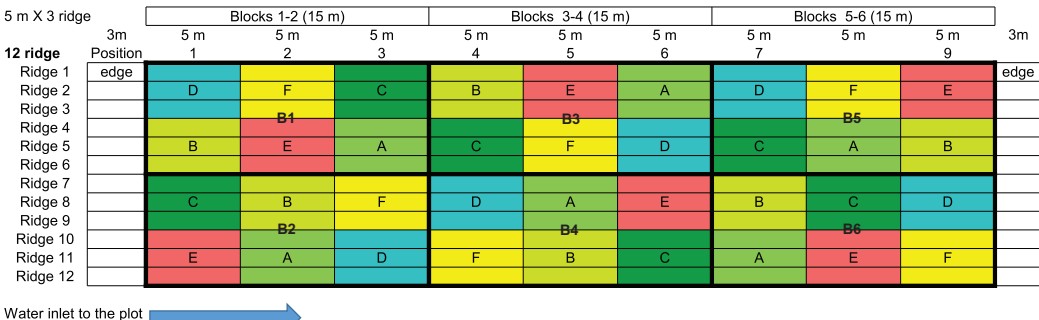

**Figure 1 Experimental design.** Treatments are noted as A: *Urtica* slurry recommended dose (RD); B: *Urtica* slurry 1/2 RD; C: *Urtica* slurry 2 × RD; D: *Urtica* + *Equisetum* slurry; E: conventional foliar manure; F: control treatment. Blocks are noted as B1, B2, B3, B4, B5, and B6.

amino acids, of which asparagine and arginine accounted for up to 80%. These nitrogen compounds were stored mainly in roots and rhizomes (*Rosnitschek-Schimmel, 1985*).

The chemical composition of several green solid manure, including nettle, was compared by *Sorensen & Thorup-Kristensen (2011)*. They concluded that nettle manure had high concentrations of boron (B), and that the low C:N ratios of green manures had a stronger impact on plant production than the total amount of N through solid soil applications (*Sorensen & Thorup-Kristensen, 2011*). To our knowledge, no scientific analysis of the effect of nettle slurry on horticultural crop yields with different doses has been published.

In this study, we aimed to analyze the possible effect of nettle slurry on potato yields produced by organic farming. To achieve this main objective, we aimed to assess: (i) the effect of nettle slurry on potato yields; (ii) the influence of slurry on potato plant size and growth parameters; (iii) its effect on chlorophyll content; (iv) its possible influence on the presence of pests and diseases. Different slurry doses were assessed in 36 plots and nine variables were measured during the crop cycle. We designed a robust randomized complete block experiment, coded for unbiased management, with many repetitions. The experiment was conducted under organic agricultural conditions.

## MATERIALS AND METHODS

### Experimental design

In this experiment, six treatments were planned; three with different *Urtica* doses (A–C) as so: one with *Urtica* and *Equisetum* slurry (D); one with conventional foliar manure (E); one control treatment (F) treated only with water (Fig. 1). The experiment was run with a randomized complete block design with six replicates per treatment and 36 plots distributed in six blocks. Each plot was 5 m long and 3 m wide (three ridges) and each plot covered 15 m² (Fig. 1). All the plants in each plot were treated. However, measurements were only taken in the three central meters of the central ridge to avoid the edge effect (Fig. S1). Plots were coded for unbiased management.

**Table 1 Main management events schedule (month/day).**

| 2/26 | 3/14 | 3/28 | 4/4 | 4/14 | 4/14 | 4/14 |
|---|---|---|---|---|---|---|
| Sowing | Irrigation (1) | Irrigation (2) | Rainfall | Re-ridge | Plot design | Measurement (1) |
| 4/14 | 4/15 | 4/16 | 4/25 | 5/3 | 5/4 | 5/23 |
| Soil samples | T. application (1) | Irrigation (3) | Irrigation (4) | Measurement (2) | T. application (2) | T. application (3) |
| 5/23 | 6/1 | 6/1 | 6/6 | 6/10 | 6/11 | 6/11 |
| Irrigation (5) | Measurement (3) | Chlorophyll samples | Irrigation (6) | Aerial biomass | Harvest | Tuber size |

**Note:**
T. application, treatment application; measurement (1) (2) and (3) corresponds to T1, T2, and T3.

**Table 2 Site soil analysis at four points A, B, C, D.**

| | Ca | | Mg | | K | | Na | | P | | Total N | | Carbonates | | NOM | | C/N | | EC | pH in W | pH in KCl |
|---|---|---|---|---|---|---|---|---|---|---|---|---|---|---|---|---|---|---|---|---|---|
| A | 5.02 | L | 0.26 | L | 1.16 | H | 0.13 | L | 148.20 | H | 0.13 | L | 42.63 | H | 2.54 | H | 11.46 | H | 385 | 8.42 | 7.84 |
| B | 5.04 | L | 0.27 | L | 1.29 | H | 0.14 | L | 82.20 | H | 0.10 | L | 34.54 | H | 2.44 | M | 14.61 | H | 294 | 8.6 | 7.92 |
| C | 5.16 | L | 0.36 | L | 0.96 | H | 0.17 | L | 184.60 | H | 0.08 | L | 35.38 | H | 2.19 | M | 16.00 | H | 449 | 8.7 | 7.95 |
| D | 5.03 | L | 0.27 | L | 0.98 | H | 0.11 | L | 121.80 | H | 0.10 | L | 35.09 | H | 2.51 | H | 14.98 | H | 295 | 8.68 | 7.91 |

**Note:**
Symbols and units are: Ca, calcium (mep/100g); Mg, magnesium (mep/100g); K, potassium (mep/100g); Na, sodium (mep/100g); P, phosphorus (mg/Kg dry soil); Total N, total nitrogen (%); Carbonates (%); NOM, natural organic matter (%); C/N, nitrogen carbon ratio; EC, soil electrical conductivity (µS); pH W, pH in water; pH in KCl.

## Experimental site

The experiment was conducted in the town of Godella in the province of Valencia, Spain (39°31′12.9″N, 0°24′34.9″W) in a calcareous silty clay loam alluvial soil from February to June 2016. The general site climate is Mediterranean oceanic, with a long-term average annual rainfall of 468 mm and an average annual air temperature of 17 °C (w.s. 39°29′N, 0°23′W 13 m a.s.l.). In 2016, the annual rainfall was 382 mm and it rained 58 mm during the experimental period. The plot was fertilized once at the beginning of the experiment before sowing. For this, horse manure (25 t/ha) was used. Subsequently, no additional soil fertilization was carried out; therefore, the use of nettle slurry was the essential fertilizer in our fertilization program. The plot was prepared and managed and ridges were built in collaboration with a local organic farmer. Eighty kilograms of the Dutch potato variety "agria" was sown for the experiment. The total cycle lasted 107 days, with five irrigation sessions (Table 1).

## Soil, water, and slurry chemical analyses

The site soil was analyzed at the start of the experiment. The experimental site was divided into four equal plots for soil sampling (A, B, C, D; Table 2). Each sample resulted from mixing five uniformly distributed soil subsamples (detail of subsamples location in Fig. S1). Soil analyses were carried out in the laboratory following standard protocols (*AOAC, 1995*). Electrical conductivity, pH, organic matter, calcium carbonate, total nitrogen, and available potassium, sodium, phosphorus, calcium, and magnesium, were determined per soil sample. For each soil parameter, high (H), medium (M) and low (L) levels were assigned according to previous studies in the same geographical area

**Table 3  Irrigation water analysis.**

| Parameter | Value (units) | Method |
|---|---|---|
| Alkalinity | 194 (mg CaCO$_3$/l) | PNT-MA/20 |
| Bicarbonates | 191 (mg/l) | PNT-MA/20 |
| Dissolved calcium | 153 (mg/l) | PNT-MA/27 |
| Carbonates | <13 (mg/l) | PNT-MA/20 |
| Riverside classification | C3-S1 | Calculation |
| Chloride | 136 (mg/l) | SM 4500 Cl B (Ed22) |
| Conductivity (20 °C) | 1,259 ($\mu$S/cm) | SM 2510 B (Ed. 22) |
| Hardness | 527 (mg CaCO$_3$/l) | PNT-MA/27 |
| Phosphate | <0.92 (mg/l) | PNT-MA/04 |
| Dissolved magnesium | 35.1 (mg/l) | PNT-MA/27 |
| Nitrate | 13.4 (mg/l NO$_3$) | PNT-MA/22 |
| pH | 8.23 ud. pH | SM 4500 H + B (ed. 22) |
| Dissolved potassium | 4.18 (mg/l) | PNT-MA/27 |
| SAR | 1.68 | PNT-MA/88 |
| Adjusted SAR | 3.87 | PNT-MA/88 |
| Dissolved sodium | 88.6 (mg/l) | PNT-MA/27 |
| Total sodium in suspension | 11.47 (mg/l) | UNE-EN 872:2006 |
| Turbidity | 12.6 (NTU) | SM 2130 B (Ed. 22) |

**Table 4  Slurry chemical analysis.**

| Slurry/parameter | pH | EC | OM (%) | OOM (%) | Ash (%) | %K$_2$O (p/V) | %P$_2$O$_5$ (p/V) | Total $N$ (%) | Protein |
|---|---|---|---|---|---|---|---|---|---|
| *Urtica* slurry | 7.92 | 1.362 | 0.001 | 0.5 | 0.07 | 0.015 | 0.0019 | 0.005 | 0.027 |
| *Equisetum* slurry | 7.66 | 1.233 | 0.007 | 0.64 | 0.095 | 0.013 | 0.0024 | 0.002 | 0.015 |

Note:
Symbols and units are EC, electrical conductivity (mS); OM, organic matter (%); OOM, oxide organic matter (%).

(*Jackson, 1976*; *Villalbi & Vidal, 1988*; *Guigou et al., 1989*; *Legaz et al., 1995*). The water analysis results were provided by *Ambitec laboratorios* (laboratory reference: 7938A) using a standard methodology (Table 3). Electrical conductivity, pH, organic matter, ash (%), K$_2$O (%), P$_2$O$_5$ (%), total N (%) and protein were determined for *Urtica* and *Equisetum* slurry (Table 4). Slurry analyses were carried out in the laboratory following standard protocols (*Horwitz, 1989*). Slurry chemical analyses were redone several times and with different batches to confirm the obtained values.

## Treatments

Commercial foliar fertilizers were used to prepare the treatment solutions. The *U. dioica* slurry (fermenting extract) and the *Equisetum hyemale* slurry came from the "*Ortiga Amiga*" trading house (Spanish company tax code (NIF) 46339215b). The base substance was produced according to regulation (EC) No. 1107/2009. The recommended *Urtica* and *Equisetum* slurry dose (RD) was 10% v/v. The conventional foliar manure was "*Isabión*" from the Syngenta trading house. The Isabión label indicates 10.9% of total nitrogen. The recommended Isabión dose was 200–300 cc/100 l of water.

Following the company recommendations, in a 10-l final volume solution the following volumes of foliar fertilizers were added to water: for treatment A (RD) 1 l; for treatment B (½ RD) 0.5 l; for treatment C (2 × RD) 2 l; for treatment D 1 l of the *Urtica* and 1 l of *Equisetum* slurry; for treatment E 25 ml of the conventional foliar manure. Control treatment F was composed of only water. Slurry was applied using a 15 l knapsack sprayer, to which we added a conical nozzle (PULMIC) to ensure a constant pressure of 2.5 bar. Treatments were applied three times throughout the crop cycle (Table 1). In the first application, plants displayed the first four true leaves (15th April); in the second, plants flowered (May 4); and in the third, plants had fully developed and came closer to the end of the cycle (May 23).

## Measurements

Crop yields, expressed as kilogram of potatoes, were determined by harvesting the central 3 m of the central ridge (Fig. S1). Three samplers worked at the same time by harvesting 1 m each to obtain three measurements per plot. The sampler effect was analyzed with no significant differences found among samplers (Fig. S2 Samplers). A total of five parameters related to plant growth and size were measured. Height, number of leaves and leaf length were measured three times throughout the plant's growth cycle (T1 is April 14, T2 is May 3, and T3 is June 1), while the number of flowers (May 3) and the final biomass (June 11) were measured once (Table 1). Each parameter was measured on five individuals from all 36 plots ($n = 180$).

A total of three leaf samples from different individuals in each plot (108 samples) were taken for the chlorophyll analyses. Sampling was performed when plants had fully developed on June 1 (Table 1). Chlorophylls were extracted with acetone protocol (*Val, Heras & Monge, 1985*) and their content was determined by absorbance spectrophotometry at wavelengths 645, 652, and 663 nm. The presence of pests and diseases was observed three times (T1, T2, and T3). Each time, five plants were noted per repetition and treatment ($n = 180$). Only Colorado potato beetle (*Leptinotarsa decemlineata* Say) and potato blight (*Phytophthora infestans* (Mont.) de Bary) were found.

## Statistical analysis

The average, standard error, skew, kurtosis, frequency distribution and density curve of the yields were assessed for each treatment. ANOVAs were used to compare the mean values between treatments and blocks. Shapiro–Wilk tests were calculated to check normality requirements. In some cases due to lack of normality, nonparametric methods were selected to compare the means among treatments and blocks by a Kruskal–Wallis rank sum test. When significant differences were found, Levene's test and eta-squared statistics were calculated to assess the homogeneity of variances and the effect size in the ANOVA, respectively. Tukey honest significant difference (HSD) was selected as the post hoc test.

To exclude the block effect (watering) from the main effect (yield), the residuals from the block ANOVAs were compared. Linear models were selected to analyze the effect of

Table 5 Average yield achieved in each treatment (kg/m).

|   | Treatments | N | Mean | se | HSD |
|---|---|---|---|---|---|
| A | *Urtica* slurry 1 RD | 18 | 2.16 | 0.1525 | a |
| B | *Urtica* slurry 1/2 RD | 18 | 2.33 | 0.1482 | a |
| C | *Urtica* slurry 2 RD | 18 | 2.27 | 0.1292 | a |
| D | *Urtica + Equisetum* slurry | 18 | 2.29 | 0.1377 | a |
| E | Conventional foliar manure | 18 | 2.05 | 0.1583 | a |
| F | Control | 18 | 2.35 | 0.2163 | a |

**Notes:**
Analysis of variance values: D$f$ = 102; $F$-value = 0.5079; $p$-value = 0.7697; HSD = 0.6554.
RD, recommended dosage; $N$, number of repetitions; se, standard error; HSD, post hoc Tukey test honestly significant difference.

numerical variables (plant size and chlorophyll variables) on yield, while Pearson correlations were calculated to assess the relationships between independent variables. A principal component analysis was used to analyze how the chlorophyll parameters were related. All the statistical analyses were done using R (*R Core Team 2017*) with some extra packages: car (*Fox & Weisberg, 2011*); plotrix (*Lemon, 2006*); ggpubr (*Kassambara, 2017*); agricolae (*de Mendiburu, 2017*); vcd (*Meyer, Zeileis & Hornik, 2006*); writexl (*Ooms, 2017*); ggplot2 (*Wickham, 2009*); readxl (*Wickham & Bryan, 2017*); plyr (*Wickham, 2011*); tidyr (*Wickham & Henry, 2017*) and knitr (*Xie, 2017*).

# RESULTS

## Soil, irrigation water, and slurry chemical composition

Soil nutrient status was homogeneous throughout the site (Table 2). Although natural organic matter contents were medium and high, the presence of high levels of carbonates and C/N ratio indicated a slow release of nitrogen to soil. Therefore, the total nitrogen value in soil was low. We obtained homogeneous pH values, which indicates strongly alkaline soil (average pH of 8.6; Table 2).

The irrigation water analyses indicated a low-medium nitrate content (Table 3). According to Riverside classification C3-S1, water salinity was medium, and therefore, appropriate to irrigate well-drained soil (*Aragues et al., 1979*). The analyses showed a high content of carbonates and soluble potassium, and a low sodium content (Table 3).

The slurry chemical analyses indicated neutral pH values (7.6–7.9) and medium electrical conductivity (1.2–1.3 mS) (Table 4). Very low levels of macronutrients (N-P-K) were obtained for both slurries. Nettle slurries from other trading houses were also analyzed with similar low levels of macronutrients (Table S1).

## Effect of nettle slurry on potato yields

Including all the treatments, 36 plots were treated and 108 yield weight measures were recorded (Fig. 1 and Fig. S3). The ANOVA for the effect of treatments on yields showed no significant differences among treatments ($p$ = 0.7697, $R^2$ = 0.0243). With the control treatment, a mean of 2.35 kg/m of potatoes were collected, while the mean was 2.16 kg/m

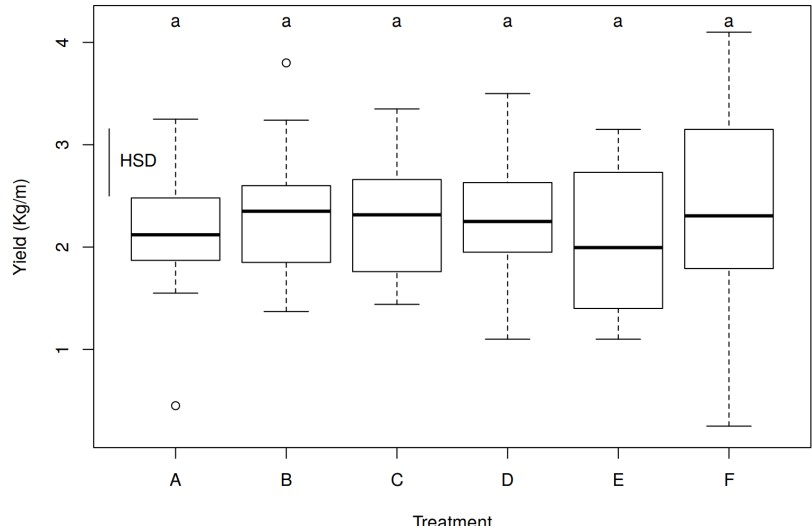

**Figure 2 Effect of treatments on yield (kg/m) with all the data (102 degrees of freedom).** Treatments are noted as (A) *Urtica* slurry RD; (B) *Urtica* slurry 1/2 RD; (C) *Urtica* slurry 2 RD; (D) *Urtica* + *Equisetum* slurry; (E) conventional foliar manure; (F) control treatment. HSD, honestly significant difference = 0.6554 kg/m. Boxes show the 25th and 75th percentiles. Lines in the boxes show the median values; columns with the same letter do not differ significantly from each other at $p \leq 0.05$ (HSD).

**Table 6 Average yield achieved in each block pair (kg/m).**

| Block pair | N | Mean | se | HSD |
|---|---|---|---|---|
| 1–2 | 36 | 1.97 | 0.0818 | b |
| 3–4 | 36 | 2.18 | 0.1060 | b |
| 5–6 | 36 | 2.57 | 0.1218 | a |

Notes:
Analysis of variance values: D$f$ = 105; *F*-value = 8.3942; *p*-value = 0.0004; HSD = 0.3514.
N, number of repetitions; se, standard error; HSD, post hoc Tukey test honestly significant difference.

with the recommended nettle treatment dose (A), and no significant differences were found (Table 5). None of the evaluated *Urtica* doses (recommended dose, half the recommended dose, double the recommended dose and *Urtica* in combination with *Equisetum*), meaning 24 independent plots, showed any significant increase in potato tuber yields compared with the control treatment (Table 5 and Fig. 2).

In order to check whether there was any overlapping factor that masked the possible effect of nettle slurry on potato yields, we analyzed whether there were any significant differences between blocks (Table S2 and Fig. S4). The ANOVA for the effect of blocks on yield showed significant differences ($p = 0.0004$, $R^2 = 0.3743$). The mean yield values increased from block 1–2 (closer to the water entry point) to block 5–6 (far away from the water entry point). Accordingly, the watering effect was analyzed. Due to the plot slope, a similar amount of water was applied to blocks 1 and 2, blocks 3 and 4, and blocks 5 and 6 (Fig. 1). Therefore, pairs of blocks were used to analyze the effect of watering (Table 6 and Fig. 3). Significant differences among watering blocks were observed

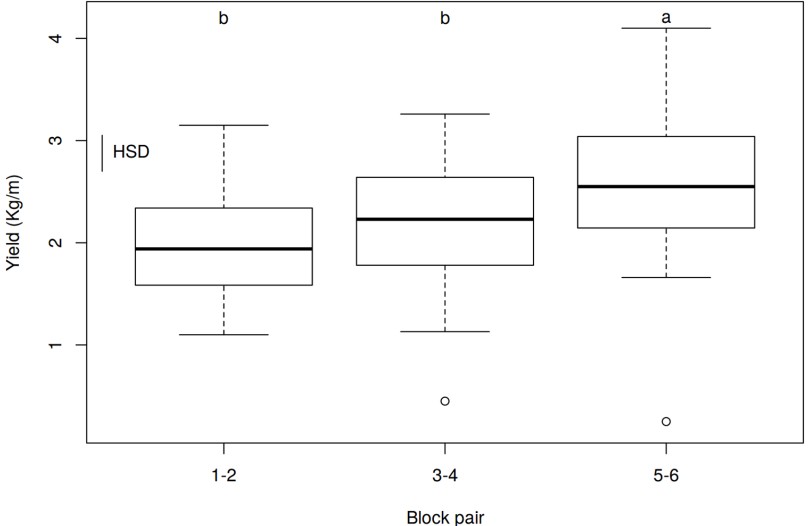

**Figure 3 Effect of watering by block pairs on yield (kg/m).** HSD, honestly significant difference = 0.3514 kg/m. Boxes show the 25th and 75th percentiles. Lines in the boxes show the median values; columns with a different letter are significantly different at $p \leq 0.05$ (HSD).

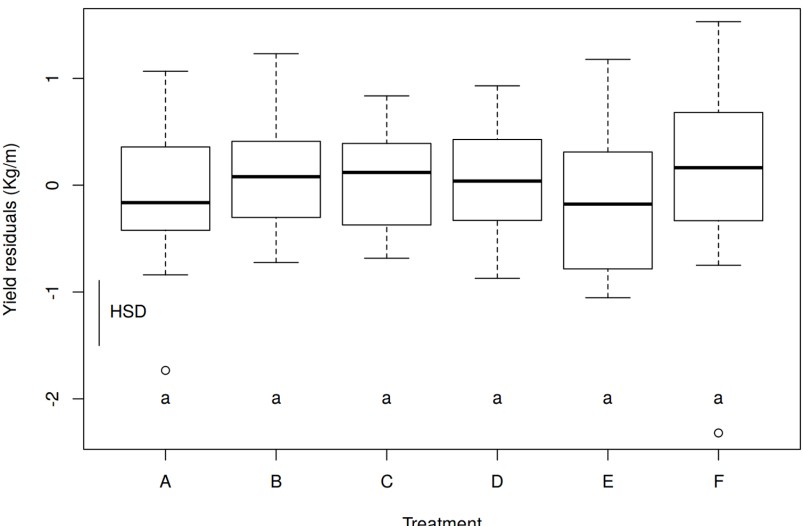

**Figure 4 Effect of treatments on yield (kg/m) with the residuals of the Block Pair ANOVAs.** Boxes show the 25th and 75th percentiles. Lines in the boxes show the median values. Df, degrees of freedom = 102; $F$-value = 0.5914; $p$-value = 0.7065; HSD, honestly significant difference = 0.6073 kg/m; columns with the same letter do not differ significantly from each other at $p \leq 0.05$ (HSD).

($p = 4.1507^{-4}$, $R^2 = 0.1378$), with a yield reduction as blocks approached the water entry point (e.g., 1.97 ± 0.08 kg/m for B1-2, and 2.57 ± 0.12 kg/m for B5-6) (Table 6).

The ANOVA for the effect of treatments on yield, excluding the watering effect, showed no significant differences among treatments ($p = 0.7065$, $R^2 = 0.0282$; Fig. 4). F value, $p$ value and HSD changed slightly by eliminating the effect of block pair, but not enough to be significant. Consequently, none of the nettle treatments resulted in significantly

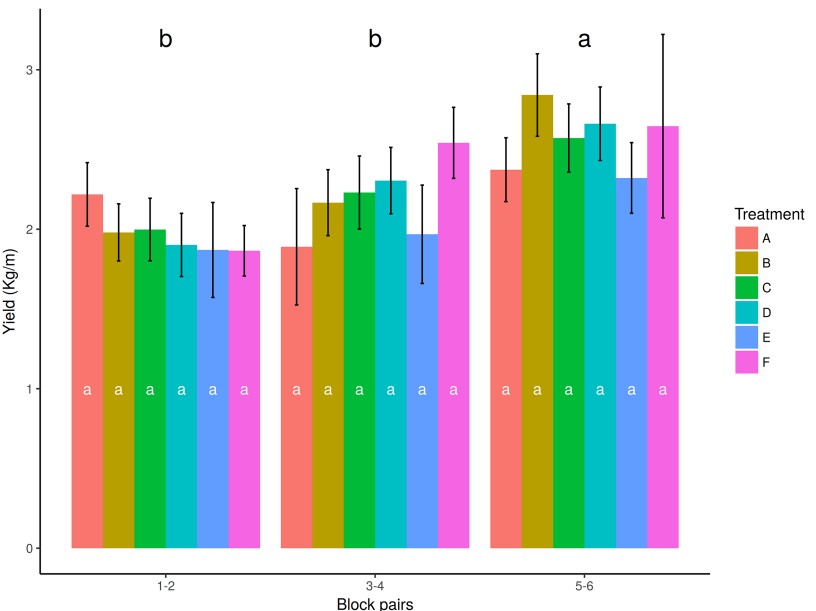

**Figure 5 Mean yield values for treatments and block pairs.** Colors represent treatments. The columns with the same letter represent values that are not significantly different at the 0.05 level of probability according to the HSD test. Error bars correspond to standard error. Treatments are noted as A: recommended *Urtica* dose; B: half the recommended dose; C: double the recommended dose; D: *Urtica* recommended dose in combination with *Equisetum*; E: conventional foliar manure; F: control. Winthin blocks, columns with the same letter do not differ significantly from each other at $p \leq 0.05$ (HSD), among blocks, different letters indicate significant differences at $p \leq 0.05$ (HSD).

increased tuber production after excluding the watering effect. When considering only tendencies (as differences were nonsignificant), the plants treated with the half dose and the control treatment yielded more tuber kilograms than those with the recommended nettle dose (Table 5).

The interaction between treatments and block pairs (Fig. 5 and Table S3) showed that a higher yield was obtained in block pair 5–6 for all the treatments. Inside the block pair, differences among treatments (Table S3) were not significant in any case.

## Influence of nettle slurry on potato growth and plant size parameters

The results of measuring the growth and size parameters (height, number of leaves, leaf length measured at T1, T2, and T3, number of flowers, and the final biomass measured once) are provided in Table 7. The ANOVA showed that the effect of treatments on these parameters was weak, with only some significant differences (Table 8). The analyses of plant height and number of flowers, both at T2, were not meaningful because the standard deviations were not homogenous among treatments (Levene's test in Table 8). The ANOVA for plant height at T3 showed significant differences ($p = 0.0051$), which indicates that the plants treated with a double RD (mean of 67.27 cm) were slightly higher at the end of the cycle than the control treatment (mean of 60.13 cm), with significant differences even though the effect size was small ($\eta^2 = 0.09$) (Fig. 6).

**Table 7 Mean and standard error of the size and growth variables for each treatment.**

| Treatments/variables | | A | | B | | C | | D | | E | | F | |
|---|---|---|---|---|---|---|---|---|---|---|---|---|---|
| | N | Mean | se | Mean | se | Mean | se | Mean | se | Mean | se | Mean | se |
| Height_T1 | 30 | 18.30 | 0.9105 | 19.13 | 0.6748 | 16.67 | 0.9288 | 18.00 | 0.6794 | 18.17 | 0.9937 | 16.20 | 0.7271 |
| Height_T2 | 30 | 51.63 | 1.2360 | 53.73 | 1.6305 | 57.97 | 1.1840 | 57.50 | 0.8699 | 48.73 | 2.2899 | 55.87 | 1.9604 |
| Height_T3 | 30 | 60.93 | 1.9350 | 63.63 | 2.0512 | 67.27 | 2.0232 | 65.57 | 1.4466 | 57.07 | 2.0396 | 60.13 | 2.4919 |
| LeavesN_T1 | 30 | 10.77 | 0.5237 | 13.17 | 0.5081 | 11.73 | 0.5393 | 13.17 | 0.5153 | 12.37 | 0.6439 | 13.40 | 0.6138 |
| LeavesN_T2 | 30 | 13.67 | 0.3269 | 15.73 | 0.3811 | 14.23 | 0.3225 | 15.30 | 0.4836 | 15.43 | 0.4062 | 15.70 | 0.4632 |
| LeavesN_T3 | 30 | 14.60 | 0.3876 | 17.17 | 0.6977 | 17.30 | 0.6149 | 16.83 | 0.7344 | 17.23 | 0.7293 | 16.73 | 0.6616 |
| LeavesL_T1 | 30 | 19.90 | 0.8475 | 21.93 | 0.6341 | 16.27 | 0.7027 | 18.90 | 0.8863 | 19.40 | 0.9416 | 16.40 | 1.0044 |
| LeavesL_T2 | 30 | 28.23 | 0.4154 | 28.47 | 0.4450 | 29.30 | 1.0397 | 27.87 | 0.7486 | 28.10 | 0.6176 | 29.93 | 0.8549 |
| LeavesL_T3 | 30 | 31.50 | 0.5427 | 34.03 | 0.5384 | 33.53 | 0.5571 | 32.93 | 0.7142 | 32.13 | 0.7095 | 32.33 | 0.7133 |
| FlowersN_T2 | 30 | 13.17 | 1.4676 | 20.23 | 0.7435 | 19.73 | 0.8495 | 18.70 | 0.9565 | 20.57 | 1.4575 | 16.80 | 1.2179 |
| AerBiom | 18 | 1.56 | 0.0353 | 1.70 | 0.1695 | 1.82 | 0.2016 | 1.62 | 0.1264 | 1.54 | 0.1442 | 1.71 | 0.1219 |

Note:
Variables and units are, Height_T1, plant height in cm at time 1; LeavesN_T1, number of leaves at time 1; LeavesL_T1, leaf length in cm at time 1; the same for T2 and T3; FlowersN_T2, number of flowers at time 2; AerBiom, aerial biomass in kg in each meter of the ridge.

**Table 8 ANOVA and post hoc Tukey test performed with the residuals of the Block Pair.**

| Size and growth variables | p | KW-p | HSD | eta2 | levene | shap | A_hsd | B_hsd | C_hsd | D_hsd | E_hsd | F_hsd |
|---|---|---|---|---|---|---|---|---|---|---|---|---|
| Height_T1 | 0.12486 | NA | 3.3791 | NA | NA | 0.0782 | a | a | a | a | a | a |
| Height_T2 | 0.0002* | 0.0029* | 6.5359 | 0.1260 | 0.000* | 0.027* | ab | ab | a | a | b | a |
| Height_T3 | 0.0051* | NA | 8.2365 | 0.091* | 0.1708 | 0.7665 | ab | ab | a | a | b | ab |
| LeavesN_T1 | 0.0062* | NA | 2.2813 | 0.088* | 0.6005 | 0.0716 | b | a | ab | a | ab | a |
| LeavesN_T2 | 0.0006* | NA | 1.6383 | 0.1155 | 0.1090 | 0.8955 | b | a | ab | ab | a | a |
| LeavesN_T3 | 0.0320* | 0.0339* | 2.6433 | 0.067* | 0.0660 | 0.004* | b | ab | a | ab | ab | ab |
| LeavesL_T1 | 0.00001* | NA | 3.4482 | 0.1591 | 0.0474 | 0.3701 | a | a | b | ab | ab | b |
| LeavesL_T2 | 0.29874 | NA | 2.9408 | NA | NA | 0.5030 | a | a | a | a | a | a |
| LeavesL_T3 | 0.05747 | NA | 2.5866 | NA | NA | 0.7173 | a | a | a | a | a | a |
| FlowersN_T2 | 0.00004* | NA | 4.6913 | 0.1467 | 0.004* | 0.0852 | b | a | a | a | a | ab |
| AerBiom | 0.73962 | NA | 0.5863 | NA | NA | 0.0012 | a | a | a | a | a | a |

Note:
p, p-value for ANOVA (*means significant differences); KW-p, Kruskal–Wallis p-value, was only calculated when residuals did not fit normal distribution (*means significant differences); HSD, post hoc Tukey test Honestly Significant Difference; eta², eta-squared statistics for the effect size in ANOVA, was only calculated when there were significant differences (*means small effect sizes); levene, levene p-value for variance verification, was only calculated when there were significant differences (*means nonhomogeneous standard deviations); shap, W of Shapiro–Wilk test of residuals (*means data did not fit normal distribution).

The ANOVA for leaf length at T1 also showed significant differences ($p = 0.00001$), which indicates that initially the leaves treated with slurry (RD and ½ RD) were larger than those from the control (e.g. 21.9 ± 0.6 cm for ½ RD, and 16.4 ± 1 cm for the control), although this effect disappeared as plants grew (T2 and T3) (Fig. 6). The differences in the number of leaves among treatments were significant at all times, but effect sizes were small ($\eta^2 = 0.09$ for T1 and $\eta^2 = 0.07$ for T3). A variation at T2 of only two leaves between the lowest (13.7) and highest (15.7) mean values was found. Nevertheless, the results of number of leaves were not consistent as the plants treated with the slurry RD had two leaves less than those of the control treatment, while the plants treated with the ½ RD

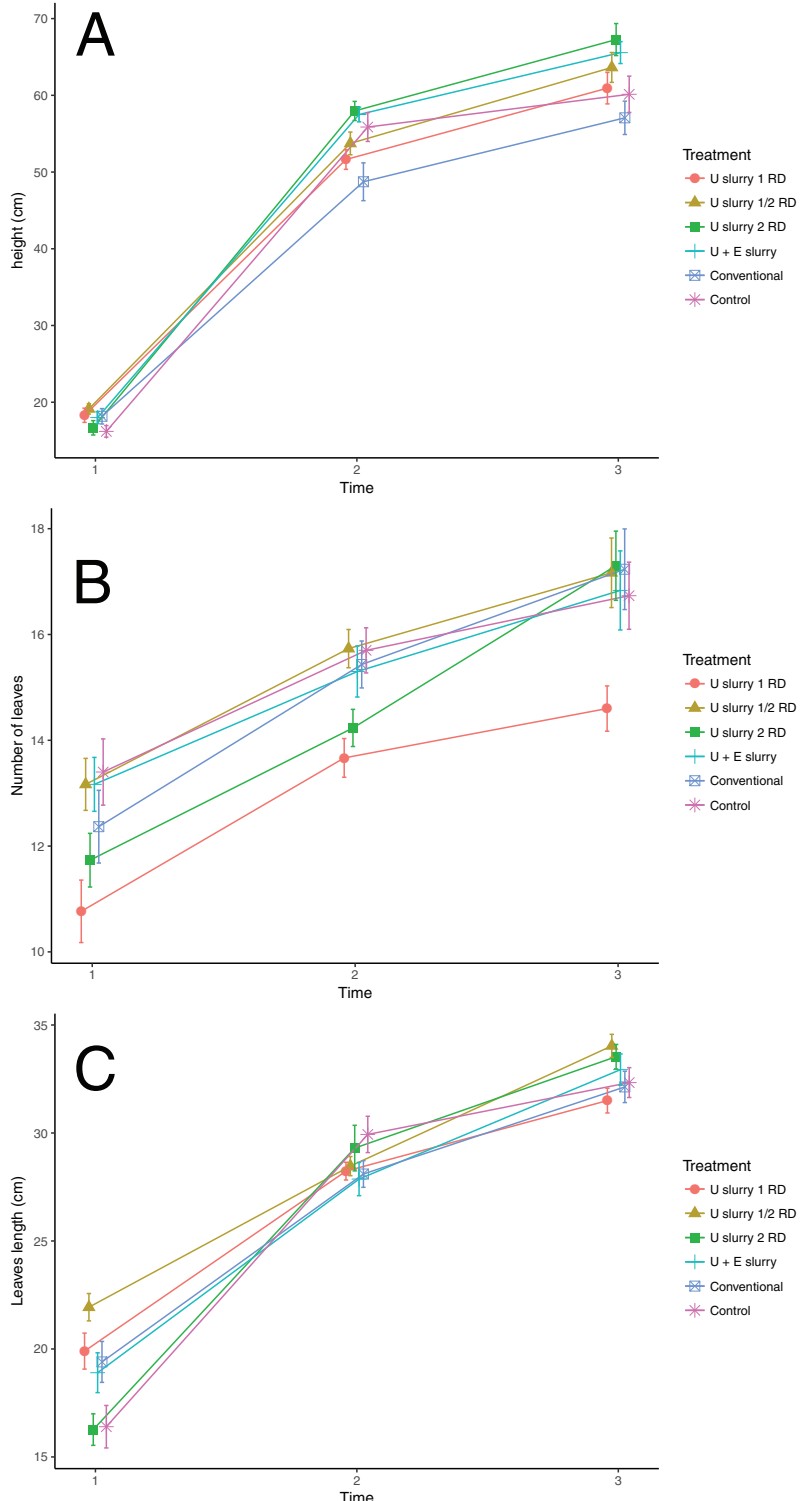

**Figure 6 Effect of treatments on the plant size variables at times 1, 2, and 3.** (A) Height; (B) Number of leaves; (C) Leaves length; Colors and symbols represent treatments. Dates (m/d) of each time were T1: 4/14; T2: 5/3; T3: 6/1. Treatments are noted as U slurry 1 RD: recommended *Urtica* dose; U slurry 1/2 RD: half the recommended dose; U slurry 2 RD: double the recommended dose; U + E slurry: *Urtica* recommended dose in combination with *Equisetum*; Conventional: conventional foliar manure; control.

**Table 9 Chlorophyll A, B and total chlorophyll content by treatment.**

| Chlorophyll variables | n | A | | B | | C | | D | | E | | F | | HSD | | | | | |
|---|---|---|---|---|---|---|---|---|---|---|---|---|---|---|---|---|---|---|---|
| | | Mean | se | Mean | se | Mean | se | Mean | se | Mean | se | Mean | se | A | B | C | D | E | F |
| A_Chlor | 108 | 0.52 | 0.04 | 0.54 | 0.04 | 0.61 | 0.05 | 0.63 | 0.03 | 0.51 | 0.03 | 0.57 | 0.05 | a | a | a | a | a | a |
| B_Chlor | 108 | 0.14 | 0.02 | 0.17 | 0.02 | 0.18 | 0.02 | 0.16 | 0.01 | 0.14 | 0.01 | 0.16 | 0.02 | a | a | a | a | a | a |
| Total_Chlor_1 | 108 | 0.19 | 0.02 | 0.21 | 0.02 | 0.23 | 0.03 | 0.21 | 0.02 | 0.18 | 0.01 | 0.21 | 0.02 | a | a | a | a | a | a |
| Total_Chlor_2 | 108 | 0.67 | 0.06 | 0.71 | 0.06 | 0.82 | 0.08 | 0.80 | 0.04 | 0.65 | 0.04 | 0.73 | 0.07 | a | a | a | a | a | a |

**Notes:**
ANOVA values are in A_Chlor $p$-value = 0.35; HSD = 0.18; B_Chlor $p$-value = 0.46; HSD = 0.08; Total_Chlor_1 $p$-value = 0.48; HSD = 0.09; Total_Chlor_2 $p$-value = 0.32; HSD = 0.26.

A, recommended *Urtica* dose; B, half the recommended dose; C, double the recommended dose; D, *Urtica* recommended dose in combination with *Equisetum*; E, conventional foliar manure; F, control; A_Chlor, A chlorophyll; B_Chlor, B chlorophyll; Total_Chlor_1, total chlorophyll 1; Total_Chlor_2, total chlorophyll 2; Unit is chlorophyll milligram per gram of fresh plant; N, number of repetitions; se, standard error; HSD, post hoc Tukey test honestly significant difference.

had the same number of leaves as the control treatment (15.7). Aerial biomass was also evaluated at the end of the cycle, and no significant differences among treatments were found (Table 8).

## Influence of treatments on chlorophyll content

Ten days before harvest, three leaves per plot were collected for chlorophyll determination purposes ($n = 108$). A chlorophyll, B chlorophyll, total chlorophyll 1 and total chlorophyll 2 were calculated with the absorbance data. No significant differences in the leaf chlorophyll content among treatments were observed (total chlorophyll 2 $p$ value = 0.3186) (Table 9). The chlorophyll parameters correlated highly (Fig. S5), therefore total chlorophyll 2, which was nearly parallel to Principal Component 1, was selected to analyze the effects of chlorophylls (Fig. S6). The plants treated with double RD had the highest total chlorophyll 2 content (mean value of 0.82 mg/g), but no significant differences compared with other treatments were found (Table 9).

## Effect on pests and presence of diseases

Pests and presence of diseases were recorded at three different times and in five plants from each plot. The Colorado beetle *L. decemlineata* appeared at T2 and slightly decreased at T3 (Fig. 7), whereas *P. infestans* appeared only at T3 for all treatments and water regimes, except for treatment E (Fig. 8). The Chi-square tests for pest occurrence were done to compare treatments. Nevertheless, there were not enough expected values for the pests present analyses, which indicates that it was not possible to correctly assess significant differences among treatments. Similar small amounts of pests were observed in all the treatments, and low levels of pests present indicated that no treatment significantly increased the appearance of pests.

## Correlations among the evaluated variables

Plant size and the chlorophyll parameters did not affect yield, while some of these parameters were affected by treatments or watering. Chlorophyll content was not affected by watering blocks ($p > 0.05$), but correlated positively with biomass ($R = 0.54$), height

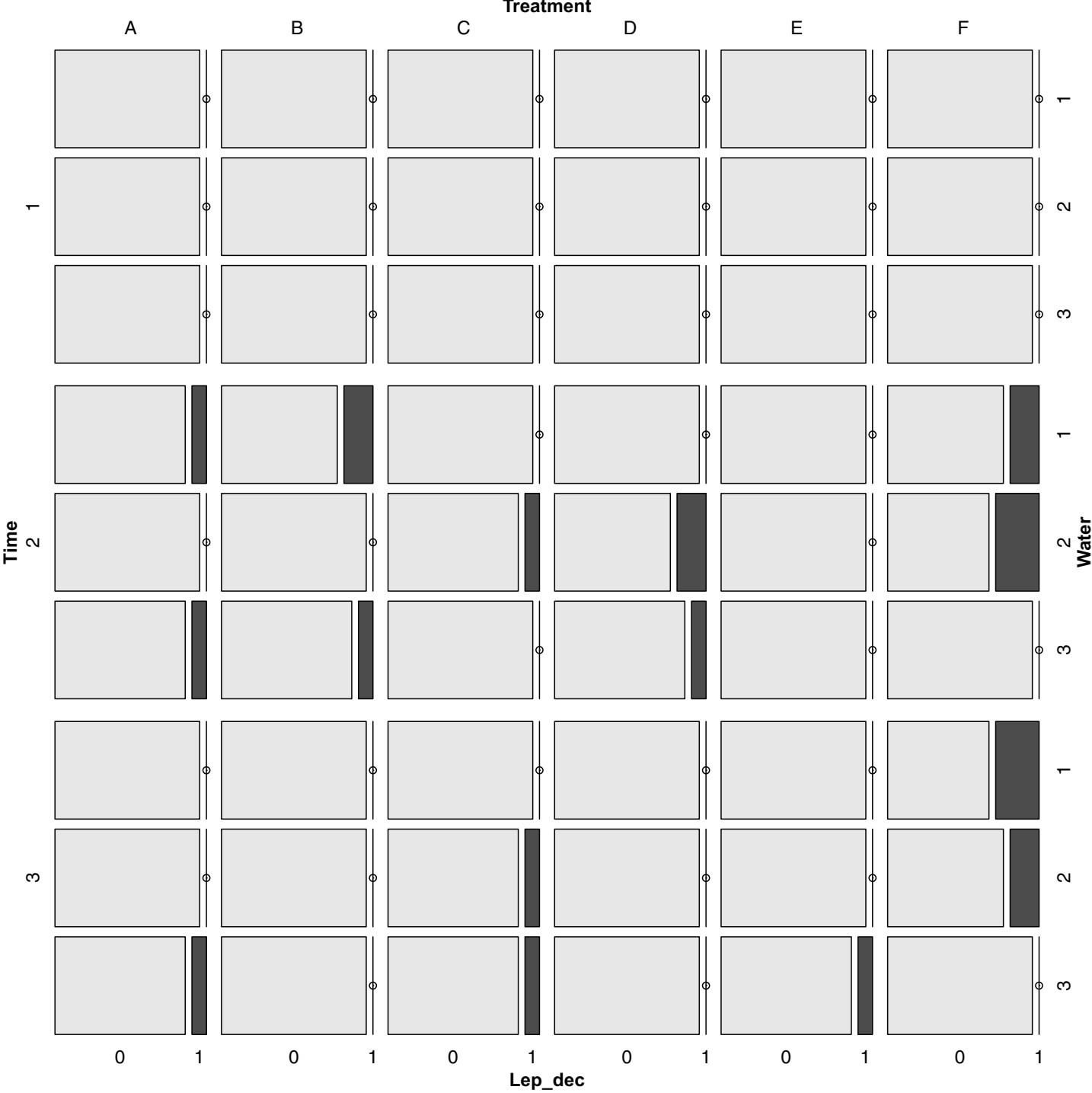

**Figure 7  Mosaic plot for the presence of *Leptinotarsa decemlineata* for different times, treatments and watering regimes.** Time dates (m/d) were T1: 4/14; T2: 5/3; T3: 6/1. Treatments are noted as (A) recommended *Urtica* dose; (B) half the recommended dose; (C) double the recommended dose; (D) *Urtica* recommended dose in combination with *Equisetum*; (E) Conventional foliar manure; (F) control. Watering regimes were 1: blocks 1 and 2; 2: blocks 3 and 4; 3: blocks 5 and 6.

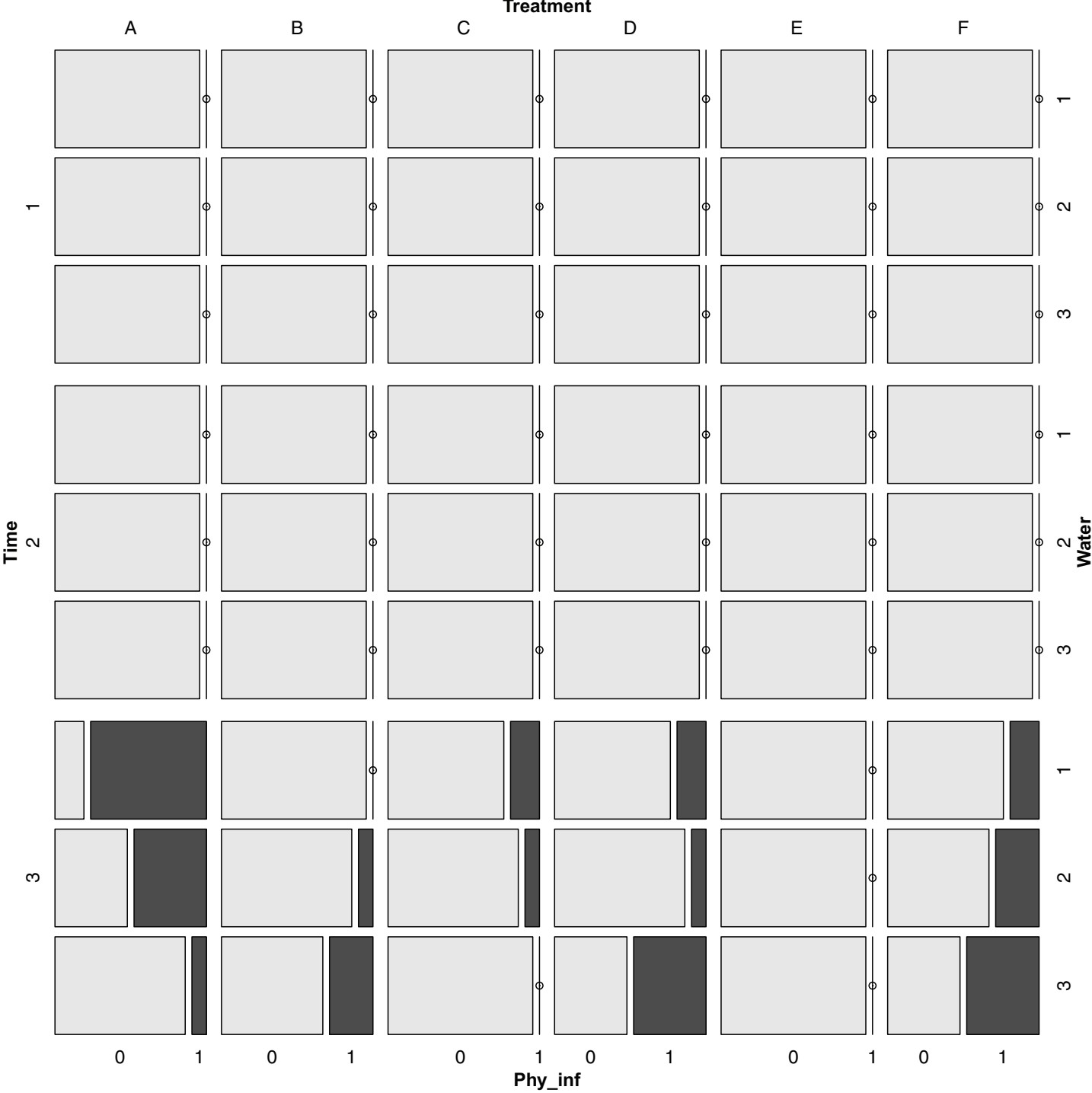

**Figure 8 Mosaic plot for the presence of *Phytophthora infestans* for different times, treatments and watering regimes.** Time dates (m/d) were T1: 4/14; T2: 5/3; T3: 6/1. Treatments are noted as (A) recommended *Urtica* dose; (B) half the recommended dose; (C) double the recommended dose; (D) *Urtica* recommended dose in combination with *Equisetum*; (E) Conventional foliar manure; (F) control. Watering regimes were 1: blocks 1 and 2; 2: blocks 3 and 4; 3: blocks 5 and 6.

($R = 0.62$) and number of leaves ($R = 0.47$) at T3, and to a lesser extent at T2. These findings indicate that large plants had a higher chlorophyll content.

Pests and diseases did not correlate with one another, nor with yield ($R = 0.06$ for beetle; $R = 0.18$ for blight) or watering blocks. *P. infestans* correlated negatively with plant height ($R = -0.52$), number of leaves ($R = -0.57$) and number of flowers ($R = -0.44$), which indicates that this disease appeared more frequently in small or weak plants. The plant size parameters correlated highly with one another (i.e., $R = 0.70$ for height T3—number of leaves T3), but surprisingly did not relate with yield or watering blocks ($R^2$ between 0.01–0.001 and $p > 0.05$).

## DISCUSSION

Nettle slurry foliar fertilizer treatments (including *Urtica* combined with *Equisetum*) did not have any effect on yield, chlorophyll content or the presence of pest and diseases of potato organic crops grown under these field experimental conditions. The results showed slightly improved plant appearance with higher plants at the end of the growth cycle.

Quite often several positive effects are attributed to nettle slurry, but they are based on very little scientific evidence. One of the main attributed effects is its ability to stimulate plant growth and to, therefore, improve yield due to its higher nutrient concentrations (nitrogen and others). Nitrogen is a major element in plants and is assimilated in free amino acids, proteins, and other nitrogenous compounds that are related to growth and development (*Ruamrungsri et al., 2010*). In nettle, the most important nitrogen is stored in roots and rhizomes (*Rosnitschek-Schimmel, 1985*). The chemical analyses performed during this experiment showed a very low total nitrogen content of the *Urtica* slurry (0.005%) and the *Equisetum* slurry (0.002%). This result was confirmed several times in different batches. *Sorensen & Thorup-Kristensen (2011)* obtained between 2.2% and 3.3% of nitrogen [N] in a nettle dry chopped solid green manure. Therefore, nettle solid green manure probably has more nitrogen than liquid slurries. Moreover within a plant species, the chemical composition of green manure is influenced by the developmental stage (*Sorensen & Thorup-Kristensen, 2011*). The concentration of most nutrients usually decreases during plant ontogeny (*Kirchmann & Bergqvist, 1989*; *Sorensen, 2000*) due to a dilution effect (*Jarrell & Beverly, 1981*). Consequently, the nettle slurry total content of N and other nutrients may vary depending on many factors. In addition, better labeling and more frequent chemical analyses of commercial products would be desirable.

*Sorensen & Thorup-Kristensen (2011)* found that when applied to soil, differences in the effect of solid green manures were not due to the total amount of N applied, but to N availability and to the carbon-to-nitrogen (C:N) ratio. The green manure C:N ratio also varied with the developmental stage. The best responses on cauliflower, kale, leek, and celery are observed when applied to soil green manures with low C:N ratios (*Sorensen & Thorup-Kristensen, 2011*).

In nettle, the most important nitrogen compounds are free amino acids, of which asparagine and arginine consist up to 80% (*Rosnitschek-Schimmel, 1985*). No studies have been found on the chemical type of nitrogen present in nettle slurry and how potato leaves

absorb this N. Assimilation of ammonium and nitrate by potato plants from soil or culture solution has been studied many times in the past (*Street, Kenyon & Watson, 1946*), but absorption through potato leaves when nitrogen is applied as a foliar fertilizer has not yet been studied.

Nevertheless, nitrogen might not be the only key. Micronutrients can improve the efficient use of macronutrients (*Malakouti, 2008*). Therefore, the *Urtica* and *Equisetum* slurry could hypothetically improve potato yields due to the supplied micronutrients, which was not the case. One of the most efficient foliar natural fertilizers is seaweed (*Dhargalkar & Pereira, 2005*; *Akila & Jeyadoss, 2010*; *Asma, Hiba & Laurence, 2013*), which has been recently tested on potato (*Pramanick et al., 2017*). Seaweed (*Kappaphycus alvarezii* (Doty) Doty *ex* Silva) foliar extract application, combined with a varied dose of soil fertilizers (50%, 75% and 100% RDF, where RDF = 200:150:150 kg/ha of N/$P_2O_5$/ $K_2O$, respectively) improved the plant height, yield, and chlorophyll content of potato plants (*Pramanick et al., 2017*). In this case, the authors pointed out the fact that K-sap (*Kappaphycus* extract) is a rich source of several primary nutrients, like potassium and phosphorus, of secondary nutrients like calcium and magnesium, and also trace elements like zinc, copper, iron, and manganese. As a broad source of potassium, K-sap helped in the translocation of photosynthates to tubers (*Zodape et al., 2010*). These results highlight how foliar and soil fertilization can be related, as well as the importance of other nutrients other than only nitrogen. In our experiment, the conventional foliar manure treatment, which had only a high nitrogen content, but no phosphorus or potassium, did not show any significant differences with the control treatment (only a small dose of 0.25% v/v was tested). This result could indicate limited nitrogen absorption ability through potato leaves and the need for other elements to be present like phosphorus or potassium.

Another attributed effect of nettle slurry is its ability to stimulate microbial activity on soil, but once again, very little evidence is available. This effect would be more related with soil applications (not assessed in these experiments), to the carbon, nitrogen, and other elements supplied with slurry, and to its own microbial load. No data on microbial load or the microbial activity in relation with nettle slurry have been found. Nettle slurry is also commonly used in soil applications. Perhaps in these cases, an assumed increase in microbial activity would improve plant fertilization, which has not yet been tested.

The ability to increase chlorophyll content is also attributed to nettle slurry. In this experiment, nettle slurry treatments did not significantly increase chlorophyll content. Nevertheless, the highest chlorophyll content was found in the double dose treatment, but the difference was not significant. This result, together with only a minor improvement in plant height with the double dose treatment, might suggest minimum crop improvement which, in our experimental conditions, was not enough to improve yields.

Finally, slurry is supposed to improve pest and disease prevention due to an increase in epidermal cell walls thickness. Very few studies about the effect of nettle slurry on pest and disease were found. *Bozsik (1996)* studied the aphicidal efficiency of cold water and

fermenting nettle slurries, which had a very low or nonsignificant effect on aphids like *H. pruni*, *C. ribis* or *A. spiraephaga* (*Bozsik, 1996*). In agreement with these studies, presence of pests and disease gave no significant differences among treatments.

## CONCLUSION

The use of nettle slurry as a foliar fertilizer was assessed for the first time. Under these field experimental conditions, nettle slurry (including one treatment with *Urtica* in combination with *Equisetum*) had no significant effects on yield, chlorophyll content or presence of pests and diseases in potato organic crops. We achieved a slight increase in plant growth, but it had no consequences on yield. Very low levels of nitrogen, phosphorus, and potassium were obtained for the *Urtica* and *Equisetu*m slurry. Very few studies are available on nettle slurry and, consequently, a lot of information is lacking. The chemical composition of fermented liquid slurry and its variability has been scarcely studied. The use of either slurry as a foliar fertilizer on other horticultural crops or liquid slurry in soil applications has not been assessed. Therefore, many more studies are needed to unravel whether nettle slurry is useful or not for improving horticultural organic crops, and which mechanisms are involved.

## ACKNOWLEDGEMENTS

The authors thank the organic farmer Francisco Morcillo for his collaboration in the management of the plot.

### Funding

The authors received no funding for this work.

### Competing Interests

The authors declare that they have no competing interests.

### Author Contributions

- Alfonso Garmendia conceived and designed the experiments, performed the experiments, analyzed the data, contributed reagents/materials/analysis tools, prepared figures and/or tables, authored or reviewed drafts of the paper, approved the final draft.
- María Dolores Raigón analyzed the data, contributed reagents/materials/analysis tools, authored or reviewed drafts of the paper, approved the final draft.
- Olmo Marques conceived and designed the experiments, performed the experiments, authored or reviewed drafts of the paper, approved the final draft.
- María Ferriol performed the experiments, authored or reviewed drafts of the paper, approved the final draft.
- Jorge Royo performed the experiments, authored or reviewed drafts of the paper, approved the final draft.

- Hugo Merle conceived and designed the experiments, performed the experiments, analyzed the data, contributed reagents/materials/analysis tools, prepared figures and/or tables, authored or reviewed drafts of the paper, approved the final draft.

## Data Availability

The raw data and code are provided as a Supplemental File.

## Supplemental Information

Supplemental information for this article can be found online at http://dx.doi.org/10.7717/peerj.4729#supplemental-information.

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
