# Peer review of "Effects of nettle slurry (Urtica dioica L.) used as foliar fertilizer on potato (Solanum tuberosum L.) yield and plant growth"

_PeerJ, doi:10.7717/peerj.4729_

## Round 0.1 · original submission · Minor Revisions

We have now secured 3 reviews for you manuscript who agreed that the manuscript requires minor revisions before it can be accepted for publication. Kindly attend to the comments at your earliest convenience.

·

Basic reporting

the article is well written.
L38: replace significantly with significant
L 91 : the use of the world “real is ambiguous. Do you mean certified?
In introduction add work by Virgilio et al 2014 "the potential of stinging nettle(Urtica dioicaL.) as a crop with multiple uses" who extensively described the chemical composition of Nettle.
Figure 6: the use of the same letters to designate parameters and treatments is confusing. Please change the designation. For instance you can use Roman Numerals I, II, III for parameters or better to use parameters names: Height, Nb of leaves, Leaves length.
Figure 7: the names on figure need to be fully written in the legend
The use of the letter a in this figure is confusing. We do not know if it means no difference between treatments or no difference between parameters
Figures 8 and 9: legends are incomplete: numbers and letters need to be clearly corresponded with treatments, dates and water regimes.
Table 2: legend is repeated twice (in title and under the table). Chose a way and keep it consistent with other tables. Explain where points A, B, C, and D were to be found on the experimental site.
Table 5 title should be: “Average yield achieved in each treatment”. The information on statistical tests should be found in legend.
No need for all these columns in this table. Leave only the first 5 columns (A, treatment, N, Mean, HSD, se).
Table 6: same remarks as for table 5
Table 7 and 9: I do not see a necessity for this table. What is mentioned in the text is enough
Table 19: is repeating same information as figure 7. Keep only one of them. If you choose to present it in a table, than the title needs to be changed to: “Chlorophyll A, B and total Chlorophyll content by treatment”. Keep details on statistical tests for the legend of the table.

Experimental design

In material and methods: describe agriculture management of the plots. Detail your fertilization program: Was any soil fertilizer (such as compost) used? Or did you rely only on foliar fertilization? Was the use of nettle slurry complementary or the essential fertilizer in your fertilization program?

Validity of the findings

No comment

Additional comments

A wide range of biologically based products are now available on the market that claim to boost crop growth and help plants withstand many plant diseases. However, there are few independent, scientifically-based studies to validate the efficacy of some of these products. This is one of them.
Manuscript well written,
solid and robust experiment,
Complete statistical analyses.

·

Basic reporting

The manuscript is clear and well written. the literature is well expanded, the figures are clear and many. However, the total volume is too large for a published paper in a journal. The manuscript is closer to a thesis than a paper, which should be taken into consideration in order to make it a concise and easier to follow.
The parameters measured are many and could be reduced. For instance there is no need to measure Chlorophyll A and B if the hypothesis is the yield and growth effect of nettle slurry. Yield and growth data are sufficient to answer that question. The discussion section can be more suitable for such a detail, or even another paper altogether.

Experimental design

The design is acceptable, and is pertinent to the question. The research question is clearly formulated, and answers the lacking area on use of this type of slurry. The investigation however is too expansive and I mentioned previously.

One note regarding your soil analysis: your results showed a low nitrogen level in the soil, the methodology should clearly state if nitrogen fertilizers were applied or not to correct the deficiency you reported. The effect of nettle slurry if present would probably be very subtle, the growth effect of nitrogen or lack thereof will be the dominant effect since nitrogen is the one of the more important determinants of yield. If you added nitrogen, please do mention it, and if not, then the question would be is foliar fertilization with a low nitrogen source even detectable in a low nitrogen soil.

Validity of the findings

Refer to the section above.
Data analysis is well presented and the methods are sound.
The conclusion is relevant to the discussed parameters.

Additional comments

The work is detailed and accurate. the design is well formulated and executed, however the scope is too expansive as mentioned earlier, and many parameters can be left out, to reduce the size and benefit the readers and saving time.
The remarks on nitrogen are important to take into consideration.

Reviewer 3 ·

Basic reporting

Some more references are needed in the introduction - see General Comments

Experimental design

no comment

Validity of the findings

no comment but see General comment for a few details

Additional comments

I am aware that nettle slurry has been proposed as fertilizer and pest control in organic farming but I was not aware of any work proving its effectiveness. Therefore, I find this work highly interesting and much needed.
The paper looks fine and should be published with minor revisions.

A few notes
The authors might spend a few words addressing a couple of important issues 1) cost of such treatment (how it may impact on the farm economy), 2) scale, that is… how much land it’d take to produce the amount of treatment to be provided to an ha of organic crops (ha/ha), that will expand the amount of land that organic farming will require to produce an unit of food

Line 38-40 “and their relation with agriculture (Michelsen, 2001).”
You may provide some more recent literature as well. Michelsen (2001) is now a nearly 20 years old paper, which is the situation today? (in general I’d prefer to refer to a few references, not just one, that would better help to frame a statement/claim and may help to reduce presenting biased narratives coming from the view of a single author)

Line 40-42 “ Organic food consumption is associated with health beliefs and subjective well-being, which involves higher market values and demand (Apaolaza et al., 2018)”
There are review works that indicate that organic food is also associate with important benefits for human and environmental health
See for example
https://www.sciencedirect.com/science/article/pii/S0929139317302573
https://onlinelibrary.wiley.com/doi/abs/10.1002/jsfa.6578
https://onlinelibrary.wiley.com/doi/abs/10.1002/jsfa.6836


Line 42-43 “Moreover, in the next few years, agriculture will be pushed to become more sustainable as a global response to climate change.”
This statement seems a bit on the air… is organic farming an answer? In which terms? The authors should be aware that some scholars are critical concerning the sustainability of organic farming as, they claim, its low yield would force cropping a larger amount of land.
Of course, the paper is dealing with a specific issue, therefore the authors are not required to prove a review on the matter. Anyway, providing a few more information may help to improve the introduction.

Line 111 “ annual rainfall was 381.8 mm and it rained 58.5 mm” you may round those values

Line 351-361 “The chemical analyses performed during this experiment showed a very low total nitrogen content of the Urtica slurry (0.005%) and the Equisetum slurry (0.002)”
The authors may explain (referring in particular for the case of on farm self-made products) whether the place where plants growth/are harvested may significantly affect, or not, plant composition (concerning the effect tested in the paper). Could we expect important differences to occur comparing industrial and self-made products?

---

## Round 0.2 · accepted · Accept

Dear Hugo,
Thank you for your submission to PeerJ.
Your answer to the reviewers comments are satisfactory. I can see that the manuscript was improved and you have incorporated all the revisions requested in the revised manuscript. As such, your manuscript will be recommended for publication.

#